# Comparing Objective Functions for Segmentation and Detection of Microaneurysms in Retinal Images

**Jakob K. H. Andersen**[1,3]                                                    JKHA@MMMI.SDU.DK

**Jakob Grauslund**[2,3]                                          JAKOB.GRAUSLUND@RSYD.DK

**Thiusius R. Savarimuthu**[1]                                              TRS@MMMI.SDU.DK

[1] *The Maersk Mc-Kinney Moeller Institute, University of Southern Denmark.*

[2] *Department of Ophthalmology, Odense University Hospital.*

[3] *Steno Diabetes Center Odense.*

## Abstract

Retinal microaneurysms (MAs) are the earliest signs of diabetic retinopathy (DR) which is the leading cause of blindness among the working aged population in the western world. Detection of MAs present a particular challenge as MA pixels account for less than 0.5% of the retinal image. In deep neural networks the learning process can be adversely affected by imbalance which introduces a bias towards the most well represented class. Recently, a number of objective functions have been proposed as alternatives to the standard Crossentropy (CE) loss in efforts to combat this problem. In this work we investigate the influence of the network objective during optimization by comparing Residual U-nets trained for segmentation of MAs in retinal images using six different objective functions; weighted and unweighted CE, Dice loss, weighted and unweighted Focal loss and Focal Tversky loss. We also perform test with the CE objective using a more complex model. Three networks with different seeds are trained for each objective function using optimized hyper-parameter settings on a dataset of 382 images with pixel level annotations for MAs. Instance level MA detection performance is evaluated with the average free response receiver operator characteristic (FROC) score calculated as the mean sensitivity at seven average false positives per image thresholds on 80 test images. The image level MA detection performance and detection of low levels of DR is evaluated with bootstrapped AUC scores on the same images and a separate test set of 1287 images. Significance test for image level detection accuracy ($\alpha = 0.05$) is performed using Cochran's Q and McNemar's test. Segmentation performance is evaluated with the average pixel precision (AP) score. For instance level detection and pixel segmentation we perform repeated measures ANOVA with Post-Hoc tests. **Results:** Losses based on the CE index perform significantly better than the Dice and Focal Tversky loss for instance level detection and pixel segmentation. The highest FROC score of 0.5448 ($\pm 0.0096$) and AP of 0.4888 ($\pm 0.0196$) is achieved using weighted CE. For all objectives excluding the Focal Tversky loss (AUC = 0.5) there is no significant difference for image level detection accuracy on the 80 image test set. The highest AUC of 0.993 (95% CI: 0.980 - 1.0) is achieved using the Focal loss. For detection of mild DR on the set of 1287 images there is a significant difference between model objectives ($p = 2.87e^{-12}$). An AUC of 0.730 (95% CI: 0.683 - 0.745 is achieved using the complex model with CE. Using the Focal Tversky objective we fail to detect any MAs on both instance and image level. **Conclusion:** Our results suggest that it is important to benchmark new losses against the CE and Focal loss functions, as we achieve similar or better results in our test using these objectives.

**Keywords:** Semantic Segmentation, Detection, Diabetic Retinopathy, Diabetes, Retinal Imaging.

## 1. Introduction

Segmentation of image features provides a good basis for disease detection in medical images. Doctors make diagnostic decisions based on the presence of diseased tissue or other anatomical changes, so it makes sense that algorithms for automatic diagnosis should do the same. Image classification with deep neural networks (DNN) suffer from a lack of interpretability which can be problematic in the context of medical image analysis and computer aided diagnosis. Hence, using semantic information for disease detection is a logical approach. DNNs work well for biomedical image segmentation, with an example being the U-net architecture (Ronneberger et al., 2015) and the several variations thereof which have since been proposed. Diabetic retinopathy (DR) is the most common micro-vascular complication of diabetes which is most common metabolic disease in world. Microaneurysms (MAs) are small changes occurring in the retinal vascularity and are the earliest sign of DR. MAs appear as small red dots in the retinal tissue and represent only a very small proportion of the retina, making them very difficult to detect. As MAs independently predict the risk of sight threatening DR (Pappuru et al., 2019), early detection is important to identify patients at risk. Presence of MAs is also used for planning screening intervals for patients (Grauslund et al., 2018). Automated retinal image analysis have been an active area of research for many years (Nrgaard and Grauslund, 2018) but has recently gained increased interest due to the improved performance of DNNs. DNNs have mainly been used to perform binary classification, e.g. non-referable versus referable DR as opposed to more fine scaled classification with multiple levels of DR disease severity. In the binary classification case, many DNNs have shown results comparable to human experts (Nielsen et al., 2019), but results also demonstrate that these methods have yet to achieve the desired diagnostic accuracy for full scale DR grading. This suggests that DNNs are able to detect the macroscopic abnormalities indicative of more severe levels of DR, but fail to recognise more subtle, microscopic lesions such as MAs which would enable them to better distinguish between DR levels. MAs are present at the lowest level of DR (Wilkinson et al., 2003) and MA detection can thus be used to predict this level of disease. The ability to detect low levels of DR is important as it enables the DNN to recognize signs of the disease before it has severe effects on patient health. MA detection can also be used to manage the referral of patients in screening settings and decrease the workload on clinicians as patients with low levels of DR might need to be screened more often compared to those showing no signs. Semantic information and visual interpretability is important for the adoption of computer assisted diagnostic methods. Intepretability is often a challenge due to the black box which is said to encapsulate DNNs (Adadi and Berrada, 2018). A simple approach to solving this issue is to use semantic image information as part of the diagnostic pipeline. In this work we demonstrate this approach by training DNNs for segmentation of retinal MAs and subsequent detection of DR based on the presence or absence of MAs in an image. This approach of using the output probability maps for classification resembles classification by simple logistic regression. The advantages of this approach is the semantic information yielded by the segmentation DNNs and the ability to adjust the model's sensitivity. Another appealing quality of this method is that it requires significantly fewer images compared to classification DNNs which generally requires thousands of images in order to converge(Nielsen et al., 2019).

## 2. Methods

We compare DNNs trained for segmentation of retinal MAs. The resulting network segmentation maps are used for detection of individual MAs as well as for image level detection. In order to optimize segmentation performance, we train several DNNs using different objective functions to determine the best objective for segmentation of these small lesions. As skip connections have been shown to improve performance of DNNs for biomedical image segmentation (Drozdzal et al., 2016), we utilize a Residual U-net (Zhang et al., 2018) for all our experiments. This is a U-net based architecture with addition of short residual skip connections in the encoder and decoder. The model consist of three residual blocks in the encoder, each with two convolution layers in the main path and a short residual skip connection with one convolution layer. After each convolution layer there is a batch normalization layer and in the first residual block a dropout layer is introduced after each of these. The decoder consists of a single residual block followed by upsampling layers symmetric to the encoder. We construct two models; a simple model where the upsampling consist of upsampling with interpolation, and a complex model where *upconvolution* is performed.

### 2.1. Data:

Models were trained using publicly available retinal images from the E-ophtha database with pixel level annotations for retinal MAs (Decencire et al., 2013). The dataset consists of 382 images of which 149 contain one or more MAs. The remaining 233 images contain no MAs. Images were randomly split into training, validation and test sets of 252, 50 and 80 images respectively with an equal proportion of MA to non-MA images in each set. The validation set was used for tuning hyperparameters of the network and objective functions. Individual MA detection as well as image level detection of MAs and pixel segmentation performance was evaluated on the 80 images in the test set. The test set consisted of 50 images without MAs and 30 images with MAs (average of seven MAs per image). Additionally, image level MA detection was performed on another set of 1287 images from the Messidor database (Decencire et al., 2014) using the adjucated ICDR grades by (Krause et al., 2017). The Messidor images represent images of DR level 0 (no MAs, n = 1017) and level 1 (n = 270). Prior to training, all images were pre-processed using a common pre-processing scheme consisting of extracting the green channel, contrast limited histogram equalization and finally cropping the image borders around the retina. The training set size was artificially increased through data augmentation by sampling 128 by 128 pixel crops from the images using a sliding window approach and subsequent augmentations consisting of gamma adjustment, flipping and flipping plus warping. This yielded a total of 83030 crops for training. At test time the same augmentations were performed on the test images and the results averaged to give the final output segmentation map.

### 2.2. Evaluation:

Individual MA detection performance was evaluated using the free response receiver operator characteristic (FROC) score which is calculated as the mean sensitivity at seven average false positive per image (FPAvg) thresholds of $\frac{1}{8}$, $\frac{1}{4}$, $\frac{1}{2}$, 1, 2, 4 and 8. MAs were counted as true positives if a single pixel of predicted MAs overlapped with MAs in the ground truth

segmentation map. Segmentation performance was evaluated as the average pixel precision (AP). We trained three networks with three different seeds for parameter initialization per objective function and treated each model as a subject to perform within-subject repeated measures ANOVA with Post-hoc Tukey test for instance level detection and pixel segmentation using the the nlme (Pinheiro et al., 2020) and emmeans (Lenth, 2020) packages in R. Image level detection was defined as correctly detecting any MAs in the images of the test sets at each threshold of the network's output probability map. If MAs were detected in an image and the ground truth image contained MAs the detection was counted as a true positive and otherwise as a false positive. 1000 rounds of bootstrap sampling was performed and predictions made with an ensemble of the three models trained per objective function in order to calculate the mean AUC score and the 95% confidence interval. To test for statistical significance of image level MA detection, images were then classified as either MA or non-MA by choosing the optimal prediction threshold of the ensemble based on classification sensitivity and specificity. Finally, using the mlxtend Python library (Raschka, 2018), the Cochrans's Q (Cochran, 1950) and Post-hoc McNemar test (McNemar, 1947) was used for testing the null hypothesis of there being no difference between the choice of objective function at a significance level of 0.05.

## 2.3. Model Objectives

Six different objective functions were used in our experiments; The Crossentropy loss (CE), Focal loss (FL) (Lin et al., 2017), Dice loss (DL) (Sudre et al., 2017) and Focal Tversky loss (FTL) (Abraham and Khan, 2018) as well as $\alpha$ weighted forms of CE ($\alpha$CE) and FL ($\alpha$FL). The standard objective for training segmentation DNNs is the average pixel-wise CE Equation (1), where $p_t$ is the probability assigned to each pixel belonging to the correct class y.

$$CE(p_t, y) = -\alpha log(p_t) \tag{1}$$

The FL Equation (2) extends on the standard CE by addition of a focusing parameter $\gamma$. The idea is to differentiate between easy and hard examples and focus learning on examples with low probabilities for y. A weighting parameter $\alpha$ can be added to both the CE and FL formulations.

$$FL(p_t, y) = -\alpha(1 - p_t)^\gamma log(p_t) \tag{2}$$

The DL can be formulated as in Equation (3) where $p_{i_c}$ is the $i^{th}$ pixel probability and $g_{i_c}$ is the corresponding ground truth pixel for class c. The objective is to maximize the overlap of the probability maps and ground truth segmentation maps. In practice, a small value $\epsilon$ is added to the numerator and denominator in order to avoid division by zero in case of *empty* segmentation maps.

$$DL(p_{i_c}, g_{i_c}) = 1 - 2 \times \frac{\sum_{n=i}^{N} p_{i_c} g_{i_c}}{\sum_{n=i}^{N} p_{i_c} + \sum_{n=i}^{N} g_{i_c}} \tag{3}$$

The FTL Equation (4) is based on the Tversky index which is a generalization of the Dice loss but with the addition of weighting parameters that allow for balancing false positive and false negative examples. $p_{i_c}$ is the probability that pixel i is of the MA class c and $p_{i_{\bar{c}}}$

is the probability that pixel i is of the non-MA class, $\bar{c}$ and the same is true for the ground truth g.

$$FTL_c(p_{i_c}, g_{i_c}) = \left( 1 - \frac{\sum_{n=i}^{N} p_{i_c} g_{i_c}}{\sum_{n=i}^{N} p_{i_c} g_{i_c} + \alpha \sum_{n=i}^{N} p_{i_{\bar{c}}} g_{i_c} + \beta \sum_{n=i}^{N} p_{i_c} g_{i_{\bar{c}}}} \right)^{\frac{1}{\gamma}} \tag{4}$$

In the FL objective, the focusing parameter $\gamma$ does not directly account for class imbalance. Rather its purpose it to focus the loss on examples with low probabilities for the correct class and put less emphasis on examples with high probabilities. In theory this works on imbalanced datasets, as the class with fewer examples is expected to be the *harder* examples which will have lower probabilities. The same mechanism is applied in the FTL, where the focus is based on the Tversky index of the misclassified examples.

## 2.4. Model Training

For each objective function and model configuration we performed hyper-parameter tuning by random search using the keras-tuner Python library on a subset of 20.000 patches from the full training set. For each set of hyperparameters we trained a maximum of 12 models, running three trails per setting. Models trained for a maximum of 50 epochs with early stopping if the validation loss did not improve for 10 epochs. For all models we searched for initial learning rate $\in [1e^{-2}, 1e^{-3}, 1e^{-4}]$, dropout probability $\in [0.0, 0.2, 0.5]$ as well as L2 regularization strength $\in [0.0, 1e^{-6}, 1e^{-7}]$. For the CE loss as well as FL we performed experiments both with and without the class balancing parameter $\alpha$. For weighted loss functions we searched for optimal settings of $\alpha$ and $\gamma$. Values of the hyperparameter search space are summarized in Table 1. Afterwards, we trained three networks with optimal hyperparameter settings for each objective function and model configuration using the full training set. We trained using a batch size of 24 and three different seeds for parameter initialization. The model parameters from the epoch with the lowest combined loss on the training and validation set was used for evaluation on the test set. Models were trained using the recently proposed rectified Adam (Liu et al., 2019) combined with Lookahead (Zhang et al., 2019) optimization algorithms, as early experimentation indicated that it helped with model convergence (although we did not perform rigorous experiments). Optimizer implementations were obtained from the Python packages keras-lookahead (Loo) and keras-rectified-adam (RAd). Models were implemented and trained using the Keras deep learning framework (Chollet et al., 2015) with Tensorflow backend on different Nvidia GPUs (GeForce GTX 1080, GeForce RTX 2080, TITAN X Pascal), each training one model at a time.

Table 1: Hyperparameter Search Space for Weighted Loss Functions

| Loss | Parameters |
|------|------------|
| $\alpha$CE | $\alpha \in [0.6, 0.7, 0.8, 0.9]$ |
| FL | $\gamma \in [2, 3, 4, 5]$ |
| $\alpha$FL | $\alpha \in [0.25, 0.5, 0.75]$, $\gamma \in [2, 3, 4, 5]$ |
| FTL | $\alpha \in [0.5, 0.6, 0.7, 0.8, 0.9]$, $\gamma \in [1, 1.33, 2, 4]$ |

## 3. Results

Table 2 shows the sensitivities at the seven FPAvg values for instance level MA detection using each objective function, as well as the more complex model. Up until an FPAvg of 1 MA the CE and FL based losses as well as CE with upconvolution perform about the same. At FPAvg values of 2 and above the $\alpha$CE, FL and CE with upconvolution begin to separate from the CE and $\alpha$FL objectives. This is also illustrated in Figure 1(a). The DL performs significantly worse than all the CE based objectives while the model completely fails to detect any MAs when trained to optimize the FTL objective. In Table 3 the average FROC scores calculated as the mean sensitivity at the seven FPAvg threshold is presented along with a clinically relevant sensitivity value at a threshold of 1.08 FPAvg (Niemeijer et al., 2010). The highest average FROC score of 0.5448 ($\pm$0.0096) and clinically relevant sensitivity value of 0.5743 ($\pm$0.0054) is achieved using the $\alpha$CE objective function in training. The FL and CE with upconvolution achieve comparable results with FROC scores of 0.5383($\pm$0.0073), 0.5375 ($\pm$0.0356) and 1.08 FPAvg sensitivity values of 0.5606 ($\pm$0.0274) and 0.5689 ($\pm$0.0178). According to the Post-Hoc tukey test there is no statistical significant difference between using any of the CE based objectives with regards to the FROC score and 1.08 FPAvg value on the E-ophtha images whereas all of them outperform the DL (p<0.001). Pixel level segmentation performance evaluated using the average precision AP

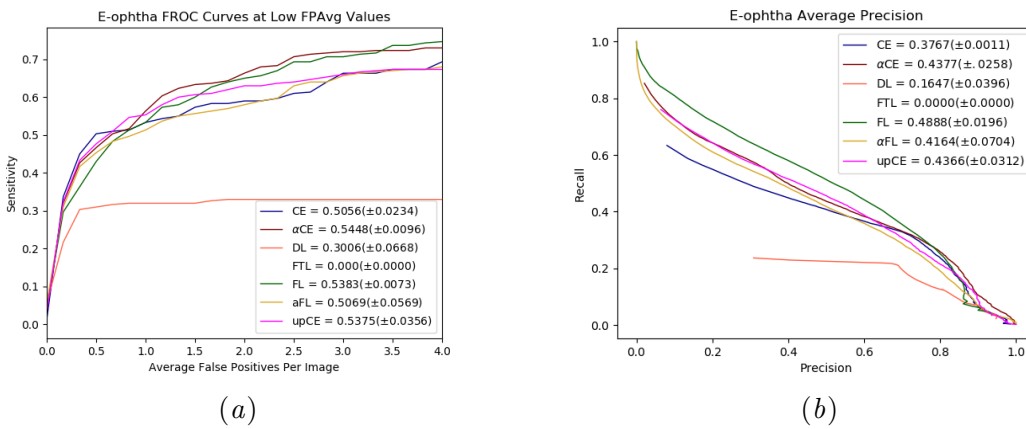

Figure 1: (a) Mean FROC curves at low FPAvg and FROC scores ($\pm sd$) and (b) precision-recall curves and average precision scores for each objective function on E-ophtha dataset.

score is presented in Figure 1(b) and Table 3. The highest AP of 0.4888 ($\pm$0.0196) is achieved using the FL objective for training but there is no significant difference between any of the CE based losses. The $\alpha$CE, CE with upconvolution, $\alpha$FL and CE achieve scores of 0.4377 ($\pm$0.0258), 0.4366 ($\pm$0.0312), 0.4164 ($\pm$0.0704) and 0.3767 ($\pm$0.01096) respectively. Using the DL the model achieves a AP of 0.1647 ($\pm$0.0396) which is significantly lower than all other objectives (p < 0.001) except for the FTL where we fail to successfully segment any MA pixels. Image level detection results on the E-ophtha and Messidor images are summarized in Table 4 using bootstrapped AUC scores with 95% confidence intervals

Table 2: Average Sensitivity Scores at Low FPAvg Thresholds on E-ophtha MA test set.

| Loss function (parameters) | $\frac{1}{8}$ | $\frac{1}{4}$ | $\frac{1}{2}$ | 1 | 2 | 4 | 8 |
|---|---|---|---|---|---|---|---|
| CE | 0.2279($\pm$0.0358) | 0.3587($\pm$0.0458) | **0.4912($\pm$0.0076)** | 0.5266($\pm$0.0116) | 0.5793($\pm$0.0235) | 0.6610($\pm$0.0456) | 0.6943($\pm$0.0325) |
| $\alpha$CE ($\alpha$=0.9) | 0.2432($\pm$0.0460) | 0.3490($\pm$0.0202) | 0.4584($\pm$0.0159) | **0.5556($\pm$0.0124)** | **0.6623($\pm$0.0261)** | 0.7458($\pm$0.0350) | 0.7995($\pm$0.0149) |
| FL ($\gamma$=5) | **0.2485($\pm$0.0825)** | **0.3377($\pm$0.0366)** | 0.4294($\pm$0.0196) | 0.5355($\pm$0.0232) | 0.6497($\pm$0.0278) | **0.7489($\pm$0.0366)** | **0.8187($\pm$0.0168)** |
| $\alpha$FL ($\gamma$=5, $\alpha$=0.25) | 0.2296($\pm$0.0602) | 0.3589($\pm$0.0994) | 0.4540($\pm$0.0482) | 0.5162($\pm$0.0398) | 0.5762($\pm$0.0513) | 0.6810($\pm$0.0708) | 0.7325($\pm$0.0513) |
| DL | 0.2162($\pm$0.0362) | 0.2629($\pm$0.0396) | 0.3110($\pm$0.0784) | 0.3219($\pm$0.0782) | 0.3308($\pm$0.0864) | 0.3308($\pm$0.0864) | 0.3308($\pm$0.0864) |
| FTL ($\gamma$=1, $\alpha$=0.7, $\beta$=0.3) | 0.0000($\pm$0.0000) | 0.0000($\pm$0.0000) | 0.0000($\pm$0.0000) | 0.0000($\pm$0.0000 | 0.0000($\pm$0.0000) | 0.0000($\pm$0.0000) | 0.0000($\pm$0.0000) |
| CE + Upconvolution | 0.2999($\pm$0.0547) | 0.3813($\pm$0.0677) | 0.4696($\pm$0.0276) | 0.5523($\pm$0.0298) | 0.6251($\pm$0.0329) | 0.6725($\pm$0.0220) | 0.7620($\pm$0.0238) |

Table 3: Average FROC, clinically relevant sensitivity scores and average pixel precision on E-ophtha MA test set.

| Loss function (parameters) | FROC Score | 1.08 FPavg Score | AP score |
|---|---|---|---|
| CE | 0.5067($\pm$0.0115) | 0.5333($\pm$0.0173) | 0.3767($\pm$0.0013) |
| $\alpha$CE ($\alpha$=0.9) | **0.5448($\pm$0.0096)** | **0.5743($\pm$0.0045)** | 0.4377($\pm$0.0258) |
| FL ($\gamma$=5) | 0.5383($\pm$0.0073) | 0.5606($\pm$0.0274) | **0.4888($\pm$0.0196)** |
| $\alpha$FL ($\gamma$=5, $\alpha$=0.25) | 0.5069($\pm$0.0569) | 0.5246($\pm$0.0441) | 0.4164($\pm$0.0704) |
| DL | 0.3006($\pm$0.0668) | 0.3219($\pm$0.0782) | 0.1647($\pm$0.0396) |
| FTL ($\gamma$=1, $\alpha$=0.7, $\beta$=0.3) | 0.0000($\pm$0.0000) | 0.0000($\pm$0.0000) | 0.000($\pm$0.0000) |
| CE + Upconvolution | 0.5375($\pm$0.0356) | 0.5689($\pm$0.0178) | 0.4366($\pm$0.0312) |

(CI) and prediction accuracy from ensemble predictions based on an optimal sensitivity and specificity prediction threshold. Models trained with either of the objective functions (excluding the FTL) are able to detect MA images with high accuracy ($> 0.87$) on the E-ophtha images. The Cochran's Q test indicate a statistical significant difference between the objective functions or model configuration (also when excluding the FTL objective) for image level detection (p = $2.07e^{-02}$) but the McNemar test with Holm-Bonferroni correction for pairwise comparison indicate no significant difference between the objective with the highest accuracy ($\alpha$FL) and the other objectives. For detection of mild DR in the Messidor images all models achieve significantly lower AUC and accuracy scores compared to the E-ophtha values. A statistical significant difference in predictive performance between the choice of objective function or model configuration is assumed on this dataset as well (p = $2.90e^{-12}$). On the Messidor images, the aCE objective has the highest accuracy and the predictive performance for detection of mild DR is significantly different from all objectives except the FL. On the Messidor dataset the standard CE loss performs significantly worse than the other objectives apart from the aFL (not including the FTL).

Table 4: Bootstrapped AUC and ensemble prediction accuracy on E-ophtha and Messidor test sets.

| Loss function | E-ophtha AUC (95% CI) | E-ophtha Accuracy | Messidor AUC (95% CI) | Messidor Accuracy |
|---|---|---|---|---|
| CE | 0.978 (0.947 - 1.0) | 0.8987 | 0.671 (0.670 - 0.731) | 0.5501 |
| $\alpha$CE($\alpha$=0.9) | 0.978 (0.948 - 0.997) | 0.8734 | 0.715 (0.683 - 0.745) | **0.6161** |
| FL($\gamma$=5) | **0.993 (0.980 - 1.0)** | 0.8987 | 0.701 (0.675 - 0.731) | 0.5990 |
| $\alpha$FL($\gamma$=5, $\alpha$=0.25) | 0.984 (0.954 - 1.0) | **0.9493** | 0.720 (0.691 - 0.753) | 0.5648 |
| DL | 0.993(0.979 - 1.0) | 0.9240 | 0.706(0.675 - 0.742) | 0.5897 |
| FTL($\gamma$=1, $\alpha$=0.7, $\beta$=0.3) | 0.500 (0.500 - 0.500) | *0.6329* | 0.500 (0.500 - 0.500) | *0.7902* |
| CE + Upconvolution | 0.977(0.940 - 1.0) | 0.9113 | **0.730 (0.683 - 0.745)** | 0.5951 |

## 4. Discussion

In this work we have demonstrated the use of DNNs for segmentation and detection of MAs in retinal images. Residual U-nets were trained to optimize six different loss functions in order to determine which objective was best suited to deal with the large class imbalance between MA and background pixels. The losses based on the CE index perform better than the Dice and Focal Tversky objectives for instance level detection of MAs. While the DL and FTL functions both have been proposed as means to deal with large class imbalance in segmentation datasets, neither of them are able to improve the performance of our networks. Even though the DL and FTL are based on overlap measures they do not lead to improved pixel level segmentation performance either, they do in-fact perform significantly worse. The weighted CE loss performs best for instance level detection with a FROC score of 0.5448 ($\pm$0.0096) although it is only the DL that performs significantly worse with a score of 0.3006 ($\pm$0.0668). While the FROC score is a good indicator of detection performance, it is potentially problematic as it can be disproportionately affected by sensitivity values at either end of the scale. E.g. their is marginal improvements in the DL sensitivities beyond $\frac{1}{2}$ FPAvg. The Reason is that the model trained using the DL produces output segmentation maps with a higher degree of *confidence* compared to the objectives based on the CE index Figure 2($a$) and Figure 2($b$). As such, the sensitivity of the model can only be adjusted in a narrow range. Of course, sensitivities above the 1.08 FPAvg threshold has little clinical relevance hence the sensitivity at this threshold is a better indicator of the true performance of a model. But as the DL also achieves lower scores here it makes no difference for the sake of comparison. Figure 3 shows an example test image with indication of correctly

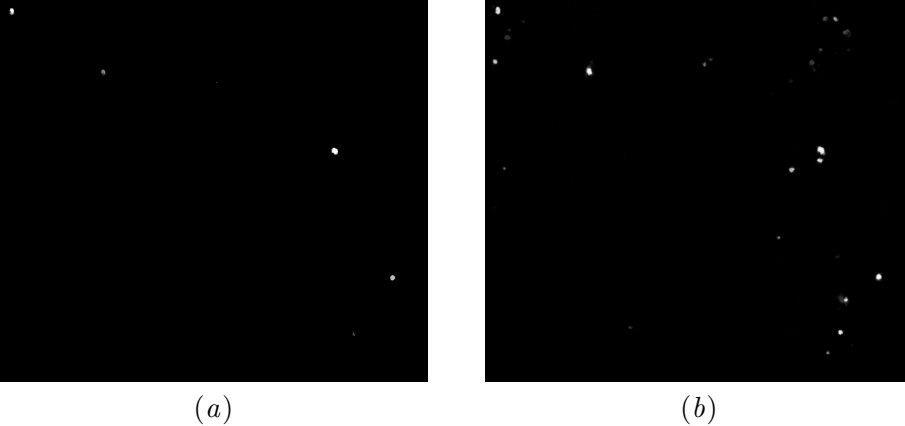

$(a)$ $\qquad\qquad\qquad\qquad\qquad$ $(b)$

Figure 2: Example crops of output probability maps from the same image: (a) Dice loss and (b) weighted Crossentropy.

and incorrectly predicted MAs from a network trained using weighted CE at a 1.08 FPAvg threshold. It is clear that some of the wrongly predicted lesions look very similar to MAs and they could in fact have been missed by the human annotator. This is a known issue when performing supervised learning and without a human comparison we cannot know the limits for how *good* it is possible for a model to be. For image level detection on the E-ophtha

data there is no significant difference between the choice of objective other than the FTL, which when used for training fails to detect any MA images. On the Messidor images the performance drops significantly for all the trained models. Even though the DL performs worse than the other objectives for instance level detection, the same is not true for image level detection. Adding a weighting as well as focusing parameter to the standard CE can lead to better performance for instance and image level detection of MAs as well as for pixel segmentation. Niether the DL og FTL lead to better, or in some cases even comparable results in any of our tests which is somewhat surprising as we expected improvements when using these objectives specifically developed to deal with highly imbalanced datasets. In (Sudre et al., 2017), they introduce the generalized Dice loss and compare the performance of this objective to a sensitivity and specificity loss and the weighted CE loss on a dataset with similar imbalance to ours. They conclude that the difference between using the different losses is minimal on datasets with moderate imbalance, but that the generalized Dice loss is more stable when the imbalance increases and they argue that the choice of loss function is crucial when training DNNs. Generally, our results indicate that the choice of objective function is important, but contrary to their results, the CE based losses proved better suited for our task. In (Abraham and Khan, 2018) they compare their Focal Tversky Loss to the DL and Tversky loss using different network architectures on two datasets. On average, they improve results by 0.062 points for Dice coefficient and 0.0255 and 0.0625 for pixel level precision and recall respectively by using the Focal Tversky objective, although in one case precision is better when using the Dice loss. The Dice coefficient is an overlap measure and they show that the FTL can improve results for this particular metric. We did not observe similar improvements for either metric used in our experiments and in fact failed to produce any form of MA detection with this objective. As they do not compare their method to the CE or FL it is unclear how using these objectives would affect their results. In the two datasets used in their experiments the lesions on average make up 4.84% and 21.4% of the pixels, which is a lot more than the 0.02% that MA pixels account for in the training set of our experiments. Recently (Kervadec et al., 2018) proposed the boundary loss and showed that it could be combined with the DL and improve performance compared to using the DL alone. As they do not make comparisons it is not known whether the boundary loss improves upon CE based losses as well and we would need to perform further experiments to test this. DNNs can struggle with generalization, meaning that performance is better on data from the same distribution compared to data from a different distribution, i.e. different population demographics, equipment setups, operators etc. in the case of medical imaging. This is also clear from the results of Table 3 comparing the mean AUC on the E-ophtha and Messidor datasets. Similar to our work, (Orlando et al., 2018) experimented with an ensemble based approach for red lesion detection in retinal fundus images by combining DNNs with hand engineered features and Random forrest. For lesion level detection on the E-ophtha dataset this method achieved a FROC score of 0.3683 when combining all three. Their DNN alone achieved a FROC score of 0.3057 which is lower than our best performing network with a score of 0.5448 ($\pm$0.0096). They also report the sensitivity at a FPAvg value of 1 which they describe as being clinically relevant. Here, their method achieves a sensitivity of 0.3680 using the combined features and the DNN alone has a sensitivity of 0.2894. In comparison, our model using the weighted CE has a sensitivity of 0.5556 ($\pm$0.0124) at this threshold. Their algorithm is evaluated on the full E-ophtha dataset and

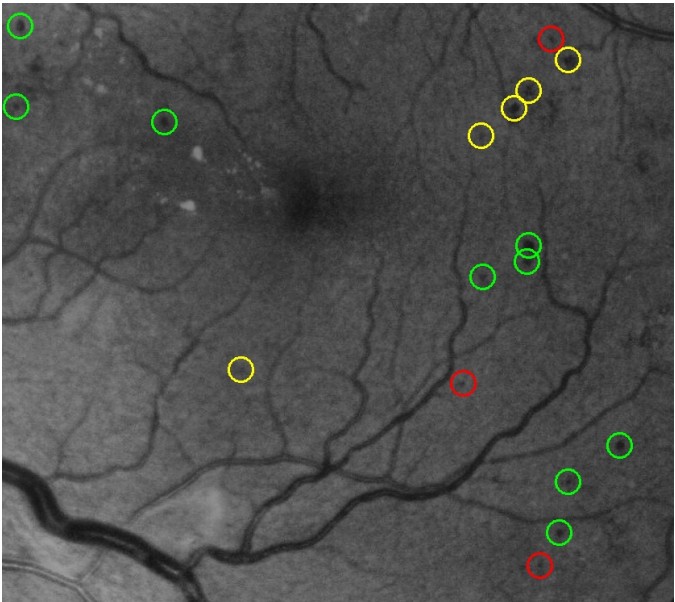

Figure 3: Preproccesed test image with indication of true positive (green), false positive (red) and false negative (yellow) predicted MAs at a clinical relevant 1.08 FPAvg threshold by network trained using weighted Crossentropy objective.

trained on images from two other databases. As in our work, they apply the algorithm to image-level detection of MAs and DR. They perform two separate tests for detection of DR in images from the Messidor database (n=1200) using the original Messidor DR grades (R0 - R3) where; R0 are images with no MAs, R1 images with between 1 and 5 MAs, R2 are images with between 5 and 15 MAs or up to 5 haemorrhages and R3 is images with 15 or more MAs or 5 or more haemorrhages or any neo-vascularization. They report an AUC of 0.8932 using the combined approach for detecting any level of DR (R0 vs R1-R3) while the DNN alone achieves AUC of 0.7912. For detection of referable DR (R0 and R1 vs R2 and R3) they report an AUC of 0.9374 using their combined approach and 0.8377 for the DNN alone. If we apply the model ensemble (CE + upconvolution) which performed best for detection of mild DR (adjucated ICDR grade 1) in the Messidor images to the same two problems using the original grades (R0-R3) our approach yields similar results with bootstrapped AUC score of 0.9005 (0.882 - 0.918) and an accuracy of 0.8185 for R0 vs. R1-R3. For R0 and R1 vs R2 and R3 our approach achieves and AUC 0.8932 (0.874 - 0.912) with an accuracy of 0.8081. On the E-ophtha dataset (Orlando et al., 2018) report an AUC of 0.9031 for their combined approach and AUC of 0.8374 for the DNN alone. Seeing as there is a difference in data used for training, comparing our method to the ensemble by Orlando et al. can be problematic. They do not train on E-ophtha data and achieve lower scores in all tests on this dataset compared to our approach, while results on the Messidor data are similar. This underlines the problem of generalization when using DNNs for medical image analysis and it is likely that the data used for training in their work is more similar to the Messidor data than to the E-ophtha data used in our experiments. A

more recent work by Chudzik et al. (Chudzik et al., 2018) train a DNN for detection of MAs using an iterative freezing approach to fine-tune a DNN using the DL objective function. They report a FROC score of 0.5620 ($\pm$0.2330) on 27 images from E-ophtha dataset but the variance of the results and the number of images used for testing renders the results unfit for comparison. (Savelli et al., 2020) perform small lesion detection using a multicontext CNN approach and report a FROC score of 0.4795 using cross validation on the E-ophtha dataset. In (Orlando et al., 2018) significant improvements are achieved by combining hand crafted features with DNN features. They employ a simple DNN architecture, and in comparison to our DNN it achieves significantly lower scores on the E-ophtha data. These results along with those by (Abraham and Khan, 2018) where deeper architectures lead to an increase in both precision and Dice coefficient indicate that more complex architectures (as our results also suggests) along with feature engineering can lead to improved results.

## 5. Conclusion

Despite the promise of using training objectives designed to deal with unbalanced data such as the DL and FTL, losses based on the Crossentropy index like weighted Crossentropy and Focal loss perform at least as well or better than these in all our experiments for lesion and image level detection as well as pixel level segmentation of small retinal MAs. While a number of new objective functions have recently been proposed and shown to improve performance for unbalanced datasets compared to the Dice loss, our results suggest that it is important to also benchmark new objectives against losses based on the Crossentropy index as we achieve the best performance in all our test using these objectives.

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
