# OpenReview forum: "Comparing Objective Functions for Segmentation and Detection of Microaneurysms in Retinal Images"
_MIDL.io/2020/Conference — MIDL 2020_

### Official Review · AnonReviewer4 · 2020-03-12
**Authors need to clarify some confusion about the effect of training data**

**Rating:** 3
**Confidence:** 4
**Recommendation:** Poster

**Summary:**

The authors face the problem of detecting retinal microaneurysms (MA) using two different approaches: (1) segmentation of the damaged area and (2) image classification. They train a residual U-net to segment the images. Then, using a specific threshold for the output probability maps, they infer the image level classification.
The imbalance of the MA pixels impedes obtaining accurate results in their detection. The aim of this work is to evaluate six different loss functions to train the network under the same conditions and determine which is the most suitable one to process unbalanced data.



**Strengths:**

The authors train a well known deep neural network architecture (residual U-net) using publicly available data and known loss functions, which makes the whole work easy to reproduce and benchmark against other approaches.
Additionally, they test their method in two kinds of datasets: (1) an independent subset of the dataset used for the training and (2) a completely independent set of images. This supports more objective conclusions and it makes possible to evaluate how general this methodology can be.



**Weaknesses:**

The authors aim to evaluate the performance of weighted cost functions when unbalanced data is processed. Contrary to what is claimed when weighted cost functions are proposed, they come up with the common cross-entropy and focal loss being the ones that provide better results.
However, for the training, during the data augmentation, they rebalance the data by including those original crops and five augmentations of them whenever the crop contains MA pixels. I would say that this is the reason why probably the weighted cost functions do not improve the accuracy results. Indeed, the authors cite the work of Sudre et al., 2017, which is similar to this one, but in their case, they avoid any data augmentation to preserve the imbalance between classes so the conclusions are different.

Another very common approach for this kind of situation, also propose by Ronneberger et al., 2015, is to define a weight map or a sampling pdf over the pixels in the image to ponderate the loss function according to each pixel and increase the weight of the unbalanced ones (pixel-wise weight loss). Is there any specific reason why the authors did not try this approach?

**Justification Of Rating:**

The authors provide an extended evaluation of the method and a deep discussion of the results obtained. However, I think their conclusions could be affected (biased) by the distribution of the training data.

**Paper Type:**

validation/application paper

**Questions To Address In The Rebuttal:**

- In the last paragraph of the introduction, it is said "Another appealing quality of this method is that it requires significantly fewer images ", could it be possible to specify why or give further details?

- How was the selection of training/validation/test images done? Did you try to get the same ratio of MA pixels in all the sets? Was it a random selection?

- The accuracy measures obtained for the segmentation are quite low, so it would help to support the results obtained if the authors could provide a measure of manual annotations variability or give more information about how difficult it is for a human to detect MA. Authors could also provide example images of the most and least challenging cases.

- Regarding the classification of mild forms of diabetic retinopathy, the authors try to detect the image levels between 1 and 5 in the Messidor dataset. Then they say "For detecting images of level 1 in the Messidor dataset defined as retinas with between 1 and 5 MAs the model trained using the CE achieves identical mean AUC of 0.8351(±0.0039)." Could it be possible to provide the results of all the cost functions?

- I would ask the authors to extend if possible, how the hyper-parameter selection and drop-out probability were initialized.

I would ask the authors to correct please, the following typos:
- "... Crossentropy loss perform at least..." --> "... Crossentropy loss performs at least..."
- " ... which is most common metabolic disease..." --> " ... which is THE most common metabolic disease..."
- "Automated retinal image analysis have..." --> "Automated retinal image analysis haS..."
- "Each levels corresponds ... " --> "Each level corresponds"
- "... individual MA detection models for each..." --> "... individual MA detection model for each..."
- "... is expected to be the harder examples..." --> "... is expected to be the harder example ..."
- "Rather its purpose it to focus ..." --> do you mean "Rather its purpose IS to focus..." ?
- "lower scores on both the E-ophtha and messidor data, these results along with those by..." --> do you mean "lower scores on both the E-ophtha and Messidor data. These results along with those by..."


**Special Issue:**

no

---

> ### Author Response · Authors · 2020-03-27
> **point by point response**
>
> - The authors aim to evaluate the performance of weighted cost functions when unbalanced data is processed. Contrary to what is claimed when weighted cost functions are proposed, they come up with the common cross-entropy and focal loss being the ones that provide better results. However, for the training, during the data augmentation, they rebalance the data by including those original crops and five augmentations of them whenever the crop contains MA pixels. I would say that this is the reason why probably the weighted cost functions do not improve the accuracy results. Indeed, the authors cite the work of Sudre et al., 2017, which is similar to this one, but in their case, they avoid any data augmentation to preserve the imbalance between classes so the conclusions are different.
>
> We would argue that even with data augmentation the data set could not be considered moderately unbalanced, as even the individual crops of 128 by 128 pixels had a high degree of foreground background imbalance. Rather, we would argue that the data augmentation could be viewed as form of regularization were perhaps a stronger regularization is being applied to the crops containing MA pixels
>
> - Another very common approach for this kind of situation, also propose by Ronneberger et al., 2015, is to define a weight map or a sampling pdf over the pixels in the image to ponderate the loss function according to each pixel and increase the weight of the unbalanced ones (pixel-wise weight loss). Is there any specific reason why the authors did not try this approach?
>
> Honestly we did not think to include this approach as we did not (strictly speaking) think it as a loss function, but in hindsight it should arguably qualify along side the weighted versions of the CE, FL and FTL. We will try to include it in the updated version of the paper if time allows it.
>
> - In the last paragraph of the introduction, it is said "Another appealing quality of this method is that it requires significantly fewer images ", could it be possible to specify why or give further details?
>
> This statement is based on Nielsen et al. wherein deep learning techniques for automatic DR detection was reviewed. In most of the literature, anywhere between a few thousands to more than 100.000 images are used in training of the algorithms.
>
> - How was the selection of training/validation/test images done? Did you try to get the same ratio of MA pixels in all the sets? Was it a random selection?
>
> The selection was random and performed separate on healthy images and images with MAs in order to ensure that the ratio of healthy to MA images was the same in the training, validation and test sets. We will specify this in the updated paper.
>
> - The accuracy measures obtained for the segmentation are quite low, so it would help to support the results obtained if the authors could provide a measure of manual annotations variability or give more information about how difficult it is for a human to detect MA. Authors could also provide example images of the most and least challenging cases.
>
>  Since the features are so small, even very small differences between the segmentation and ground truth maps leads to low segmentation scores, hence why we also evaluated the networks detection performance.  We feel that the FROC metric is a better indicator of performance. In regards to human comparison we have only encountered the one by Niemejer et al. 2009 but this is performed on another datasets where detection scores are generally much lower. We wanted to include some images and discuss the results but ultimately ran out of pages.
>
> - Regarding the classification of mild forms of diabetic retinopathy, the authors try to detect the image levels between 1 and 5 in the Messidor dataset. Then they say "For detecting images of level 1 in the Messidor dataset defined as retinas with between 1 and 5 MAs the model trained using the CE achieves identical mean AUC of 0.8351(±0.0039)." Could it be possible to provide the results of all the cost functions?
>
> We have decided to change the evaluation method such that we only test for DR level 1 for all objective functions and leave out the remaining levels and will update the table accordingly.
>
> - I would ask the authors to extend if possible, how the hyper-parameter selection and drop-out probability were initialized.
>
> We feel that this has been addressed in the methods section of the paper:
>
> ” We experimented with learning rates of 1e -3 ,1e -4 and 1e -5 and with using no dropout as well as dropout of 0.2, 0.5 and 0.7. For the Focal loss we experimented with γ of 1.5, 2, 3 and 4. For the FDL and FTL we experimented with using the value suggested in (Abraham and Khan, 2018) of 4/3 as well as 2 and 4. The weighting parameter α was set to 0.9 for the MA class and 1 - α for the background class in the αCE and αFL..."
>
> - I would ask the authors to correct please, the following typos: ...
>
> Typos will be addressed in the updated version of the paper.

---

> > ### Comment · AnonReviewer4 · 2020-04-03
> > **Response to authors**
> >
> > - "We would argue that even with data augmentation the data set could not be considered moderately unbalanced, as even the individual crops of 128 by 128 pixels had a high degree of foreground background imbalance."
> > Definitely this would depend on the pixel size of your images (which is not detailed) and the approximate size of the object you want to detect.
> >
> > - " Rather, we would argue that the data augmentation could be viewed as form of regularization were perhaps a stronger regularization is being applied to the crops containing MA pixels "
> >
> > Sorry, but I'm not sure whether I understand what you mean exactly. The problem is that this regularization might be similar to the one you would get by using a weighted loss function and this is why you do not see any improvement.
> >
> >
> >
> >
> > - "Honestly we did not think to include this approach as we did not (strictly speaking) think it as a loss function, but in hindsight it should arguably qualify along side the weighted versions of the CE, FL and FTL. We will try to include it in the updated version of the paper if time allows it. "
> >
> > For the detection task, it is true that it might not be common, but for segmentation/pixel classification it is.
> >
> >
> >
> >
> > - "This statement is based on Nielsen et al. wherein deep learning techniques for automatic DR detection was reviewed. In most of the literature, anywhere between a few thousands to more than 100.000 images are used in training of the algorithms."
> >
> > Ok, then at least, add the reference to it.
> >
> >
> >
> >
> > - "We wanted to include some images and discuss the results but ultimately ran out of pages.  "
> >
> > What about supplementary material?
> >
> >
> >
> >
> > - "We feel that this has been addressed in the methods section of the paper:"
> >
> > Reviewer 3 did also ask for further explanation. Hence, you should maybe consider rewriting it. Did you try all the possible combinations? In which layers did you use drop-out?
> > It is said "For each objective function we performed hyper-parameter search for initial learning rate setting and dropout probability". However, at the end all the models for all the cost-functions were trained with the same hyper-parameter configuration: "For our final experiments we used a learning rate of 1e-4 and dropout of 0.5 except when using FTL and FDL objectives where a dropout of 0.2 was used."

---

### Official Review · AnonReviewer3 · 2020-03-13
**A comparison of objective function for learning with large class-imbalances**

**Rating:** 2
**Confidence:** 5

**Summary:**

The authors present a comparison of different objective functions for the segmentation/detection of micro-aneurysms in retinal images. The micro-aneurysms present only a very small proportion of pixels in the input image, which may have adverse effects on learning. However, none of the objective functions that were tested were able to improve upon the cross-entropy.

**Strengths:**

The authors correctly point out that the large class-imbalance can be challenging in many problems related to segmentation/detection in medical imaging. Many different loss functions have been proposed, and the direct comparison of the performance of each of them in a particular setting is enlightening.

**Weaknesses:**


- The paper is not very well written. The long and incoherent paragraphs, make it very hard to read. Please consider separating concepts into different paragraphs.

- The choice of evaluation metrics is confusing. I don't think AUC on image level is appropriate to validate a detection/segmentation problem (also see point below). Dice or FROC-score are probably more appropriate, but I'm missing the results. The authors mention FL achieves better results for pixel segmentation, but based on what metric?

- The contribution of using the segmentation method as image level classifier is questionable. Many methods for DR classification have already been developed, and achieve expert-level performance. The classification of DR depends not just on micro aneurysms, but also on presence of hemorrhages, bright lesions, cotton wool spots.

**Detailed Comments:**

- I would suggest to replace 'tiny lesions' in the title with micro-aneurysms.
- What is the resolution (mm/pix) of the patches (crops)? How does that relate to the size of a micro-aneurysm?
- In the formulas, parameter $y$ is introduced as the correct class, but later $g$ is also used. This is confusing.
- Dice loss: it would be better to divide by the square of $p_{ic}$ and $g_{ic}$, as to obtain a dimensionless metric.

**Justification Of Rating:**

I do believe there is value in the type of comparison of different objective functions as performed in this paper. However, the presentation and experimental setup are of insufficient quality. Also, the results are not clearly communicated, it is often unclear to which specific results the authors refer.

**Paper Type:**

validation/application paper

**Questions To Address In The Rebuttal:**

- What training procedure was used? Was it the same for all objective functions?
- How did you choose hyper-parameters? And how was the best-performing setting per task assessed? Based on which metric? I believe it is important to clearly outline how each model was trained, as final performance can be strongly influenced by the training setting.

**Special Issue:**

no

---

> ### Author Response · Authors · 2020-03-27
> **point by point response**
>
> Weaknesses:
> - The paper is not very well written. The long and incoherent paragraphs, make it very hard to read. Please consider separating concepts into different paragraphs.
>
> Response:
> We appreciate the reviewers comments on the readability of the paper and will try to clean it up in order improve it.
>
> - The choice of evaluation metrics is confusing. I don't think AUC on image level is appropriate to validate a detection/segmentation problem (also see point below). Dice or FROC-score are probably more appropriate, but I'm missing the results. The authors mention FL achieves better results for pixel segmentation, but based on what metric?
>
> Response:
> As far as evaluation metric goes we use three different throughout the paper. For detection (which is based on the segmentation results by counting intersecting connected microaneurysms (MAs) between the segmentation results and ground truth segmentation maps) we use the Free response reciever operator caracteristic (FROC) score. The resulting FROC curves and FROC scores (calculated as seven average false positive MAs per image) are presented in Figure 1 (a) . We will add a table of FROC scores for each objective function with sensitivities at each threshold.  The AUC metric is applied to detection of any MAs in an image. This way we use the segmentation results to perform classification of DR/no DR. The pixel segmentation results is evaluated using the average precision metric  presented in Figure 1 (b). All the results are summarized in the abstract of the paper.
>
>
> - The contribution of using the segmentation method as image level classifier is questionable. Many methods for DR classification have already been developed, and achieve expert-level performance. The classification of DR depends not just on micro aneurysms, but also on presence of
> hemorrhages, bright lesions, cotton wool spots.
>
> Response:
> In the paper we do address the the problem of using only micro-aneurysms to detect diabetic retinopathy (DR) but we acknowledge that it has not been clearly communicated. We have decided to change the evaluation method such that we only test for DR level 1 and leave out the remaining levels due to the reasons addressed.
>
> In the relation to the other methods developed we address this in the introduction of the paper:
> ” Many automated retinal image analysis methods using DNNs perform
> binary classification, e.g. non-referable versus referable DR as opposed to more fine scaled
> classification with multiple levels of DR disease severity. In the binary classification set-
> ting, many DNNs have shown results comparable to human experts (Nielsen et al., 2019),
> but results also demonstrate that these methods have yet to achieve the desired diagnostic
> accuracy for full scale DR grading...”
>
> Questions:
>
> - What training procedure was used? Was it the same for all objective functions? - How did you choose hyper-parameters? And how was the best-performing setting per task assessed? Based on which metric? I believe it is important to clearly outline how each model was trained, as final performance can be strongly influenced by the training setting?
>
> We feel that this has been addressed in the methods section of the paper. We write:
>
> ”For each objective function we performed hyper-parameter search for initial learning rate setting and dropout probability. Afterwards, we trained three networks per setting for each of the four objective functions using three different seeds for parameter initialization using a batch size of 24.
> The model parameters from the epoch with the lowest loss on the validation set was used for
> evaluation on the test set. Models where trained with stochastic gradient descent using the Adam optimization algorithm with default parameters. We adopted the recently proposed Lookahead (Zhang et al., 2019) in the optimization step as early experimentation indicated that it helped with model convergence, although we did not perform rigorous experiments. Lookahead was implemented with its standard parameters using the python package keras-lookahead (Loo). We experimented with learning rates of 1e -3 ,1e -4 and 1e -5 and with using no dropout as well as dropout of 0.2, 0.5 and 0.7. ... ”
>
> - I would suggest to replace 'tiny lesions' in the title with micro-aneurysms.
>
> We agree with this suggestion and have decided to change the title accordingly.
>
> - What is the resolution (mm/pix) of the patches (crops)? How does that relate to the size of a micro-aneurysm?
>
> The resolution of the crops is 128 x 128 pixels.
>
> - In the formulas, parameter  is introduced as the correct class, but later  is also used. This is confusing.
>
> We will try to align the notation, but would argue that y and g represent different things, y being a single value and g a ground truth segmentation map.
>
> - Dice loss: it would be better to divide by the square of  and , as to obtain a dimensionless metric.
>
> We ran additional experiments but found that it made no discernible difference.

---

### Official Review · AnonReviewer1 · 2020-03-13
**Good experimental study with unexpected conclusions, not much novelty**

**Rating:** 3
**Confidence:** 4
**Recommendation:** Poster

**Summary:**

This paper conducts an exhaustive set of experiments on three different segmentation loss functions including weighted and unweighted variants, for the task of micro-aneurysm segmentation. The interesting bit is that the authors report findings going against what one would expect: loss functions designed for handling class imbalance largely underperform the standard cross-entropy loss.

**Strengths:**

- I believe in papers that take experimentation seriously and report results that make one re-consider the universal usefulness of widely accepted strategies, in this case for handling class-imbalance, maybe one should not take for granted that using focal loss or tuning the class weights of a cross-entropy loss will always lead to better results than using a simple CE baseline.
- The paper contains experiments training on E-ophtha and testing on both E-ophtha and a second external dataset, which is something everybody should do but few researchers do (although I have my doubts on using Messidor - see below).
- There is a very large number of experiments that seem to be doing hyperparameter optimization in a rigorous manner, and this process is described with detail.
- The provided discussion is quite rich, and not just an "I need to fill one paragraph with generic re-statement of the results"-like discussion.

**Weaknesses:**

- Obviously there is not a big deal of novelty in this paper (no new idea is presented), but if it is a considered as a validation paper I would be ok with that.
- Results are presented in a somehow hard to digest manner, and would benefit a lot from more tables. Specially, the FROC curves are ok, but it is very hard to read the actual numbers out of Figure 1's legend. It would have been much better to have a separate table with the Area under the FROC and AP.
- I am not very convinced about using Messidor to assess micro-aneurysm detection. I mean, if one takes only images from Messidor that have Diabetic Retinopathy grade 1, then it is ok, you will have micro-aneursyms there. But further grades do not imply the presence of micro-aneurysms. In Messidor-1 you can have grade 2 by having hemorrhages or grade 3 by having neo-vascularization, and not micro-aneurysms. In addition, I believe we should now be using Messidor-2, that was released several years ago and contains >1700 images with updated grading*

* Google released new grades for those images here: https://www.kaggle.com/google-brain/messidor2-dr-grades

**Justification Of Rating:**

Although this paper might probably be penalized by the lack of novelty (and I myself was tempted of choosing weak reject), I believe that if the authors follow some of the above advice to polish it a bit, it could be an interesting piece of experimental/validation research. I have also found in my own work that often baseline loss function outperform other fancy contributions from recent famous papers, and I feel it could be interesting for people to know that we should pay more attention to having a proper baseline before blindly following new "trends". In any case, there are several weaknesses to this paper that should be addressed either now, time allowing, or in future submissions of this otherwise interesting work.

**Paper Type:**

validation/application paper

**Questions To Address In The Rebuttal:**

- I think it would be important to also add the performance (FROC/AP) of other current techniques that have addressed the same problem, on the same task/datasets , just to see how far can you go if you use a specialized approach instead of a general purpose u-net. For example, the authors can find a recent method for detecting MAs in [1], although I am not asking to include that one in this work, it is just a pointer, I promise I am not the author!
- Please, please, tabulate the results given in Figure 1 so that we can read them off easily, instead of having to look into tiny legends with a magnifying glass!
- Following my above comment on using Messidor, maybe the authors could consider to test on IDRID  (https://ieee-dataport.org/open-access/indian-diabetic-retinopathy-image-dataset-idrid) or DIARET-DB1 (https://www.it.lut.fi/project/imageret/diaretdb1/)? I think both contain micro-aneurysm, probably better to use IDRID.
- As a recommendation, the authors could also look into the statistical analysis they do in [1] to take inspiration on how to do an analysis of the relevance of results in terms of statistical signficance (although hardly anyone seems to be doing this properly in most papers).
[1] A. Bria et al., A multi-context CNN ensemble for small lesion detection, Artificial Intelligence in Medicine, 2020.

- Initially I was not sure about using a segmentation approach to perform detection. However, I have been browsing the literature, and I only found approaches similar to the one here (sliding window+segmentation/classification).  Shouldn't we consider object detection approaches, YOLO-style? This is common practice in other problems that have hard imbalancing, like lung nodule detection from CT scans or micro-calcification detection from mammograms. Moreover, the Focal Loss and its variations were initially developed for Object Detection. Would it be possible to include comparison to such baseline, or at least comment on why may it be that this approach is not popular?

**Special Issue:**

no

---

> ### Author Response · Authors · 2020-03-27
> **Point by point response**
>
> Question:
> I think it would be important to also add the performance (FROC/AP) of other current techniques that have addressed the same problem, on the same task/datasets , just to see how far can you go if you use a specialized approach instead of a general purpose u-net. For example, the authors can find a recent method for detecting MAs in [1], although I am not asking to include that one in this work, it is just a pointer, I promise I am not the author!
>
> Response:
> We will  add a section in the the discussion on current state of the art on the e-ophtha and messidor datasets.
>
>
> Question:
> - Please, please, tabulate the results given in Figure 1 so that we can read them off easily, instead of having to look into tiny legends with a magnifying glass!
>
> Response:
> We will updated the results section with a table of FROC scores for each Objective function
>
> Question:
> - Following my above comment on using Messidor, maybe the authors could consider to test on IDRID  (https://ieee-dataport.org/open-access/indian-diabetic-retinopathy-image-dataset-idrid) or DIARET-DB1 (https://www.it.lut.fi/project/imageret/diaretdb1/)? I think both contain micro-aneurysm, probably better to use IDRID.
>
>
> Response:
> In the paper we address the the problem of using only micro-aneurysms to detect diabetic retinopathy (DR). As you correctly point out, higher levels of DR is defined by other lesion types  We write ”In our work we can really only use our models for detection of level 1 DR in the Messidor dataset, as the definition for higher levels also contain other imaging bio-markers such as hemorrhages and neovascularization.” and also report the results of detecting only level 1 DR using the best performing loss function (crossentropy) but aknowledge that it has not been clearly communicated. Based on your comment we have decided to change the evaluation method such that we only test for DR level 1 and leave out the remaining levels due to the reasons addressed.
>
> Furthermore, we evaluate on the updated messidor 2 data instead.  And will also added the results of testing on the IDRID segmentation data-set.
>
>
> Question:
> - As a recommendation, the authors could also look into the statistical analysis they do in [1] to take inspiration on how to do an analysis of the relevance of results in terms of statistical signficance (although hardly anyone seems to be doing this properly in most papers).
> [1] A. Bria et al., A multi-context CNN ensemble for small lesion detection, Artificial Intelligence in Medicine, 2020.
>
> Response:
> As you point out, significance test is seldom part deep learning related papers, but we will do our part to correct this by adding the proposed test for statistical significance.
>
> Question:
>
> - Initially I was not sure about using a segmentation approach to perform detection. However, I have been browsing the literature, and I only found approaches similar to the one here (sliding window+segmentation/classification).  Shouldn't we consider object detection approaches, YOLO-style? This is common practice in other problems that have hard imbalancing, like lung nodule detection from CT scans or micro-calcification detection from mammograms. Moreover, the Focal Loss and its variations were initially developed for Object Detection. Would it be possible to include comparison to such baseline, or at least comment on why may it be that this approach is not popular?
>
> Response:
> - The simplest answer to this question is that the dataset used (e-ophtha) is annotated with pixel level labels, and as such the straight forward method would be to use a segmentation network. Of cause, bounding boxes could be added around the labels in order to train an object detection network. We have decided not to add it in this paper as it is outside the scope of what we were trying to investigate, but it is definitely something which should be investigated in the future.

---

### Meta-Review · Area_Chair1 · 2020-04-05
**MetaReview of Paper174 by AreaChair1**

**Rating:** 2

**Metareview:**

The paper attempts at an in depth comparison of loss function with appropriate hyper-parameter tuning in the application of deep learning to retinal lesion detection and segmentation. It seems however overall unclear and with a lack of proper statistical analysis (which is particularly important in a validation paper with no other novel development)

**Paper Type:**

validation/application paper

**Special Issue:**

no

---

### Decision · Program_Chairs · 2020-04-11

**Decision:**

Accept

**Comment:**

Taking all information into account, it was determined that the paper was accepted based on its merit.